# A Phosphorus-Nitrogen-Carbon Synergistic Nanolayered Flame Retardant for Polystyrene

**DOI:** 10.3390/polym14102055

**Published:** 2022-05-18

**Authors:** Wen-Jie Yuan, Wei Zhao, Gang Wu, Hai-Bo Zhao

**Affiliations:** Collaborative Innovation Center for Eco-Friendly and Fire-Safety Polymeric Materials (MoE), State Key Laboratory of Polymer Materials Engineering, National Engineering Laboratory of Eco-Friendly Polymeric Materials (Sichuan), College of Chemistry, Sichuan University, Chengdu 610064, China; 2017222030124@stu.scu.edu.cn (W.-J.Y.); 2019222030132@stu.scu.edu.cn (W.Z.); wu_china_chengdu@126.com (G.W.)

**Keywords:** flame retardancy, nano-layered structure, nanocomposite, polystyrene

## Abstract

Polymers are widely used in our daily life; however, most of them are highly flammable. Once modified with flame retardants (FRs), polymers always have deteriorative properties in mechanical strength aspects. As a countermeasure, a novel unified phosphorus and nitrogen-containing organic nano-layered flame retardant (BA-MA) was synthesized by the assembly of biphenyl-4,4′-diphosphonic acid (BA) and melamine (MA), which was used as an additive flame retardant for polystyrene (PS) resin. The chemical structure and morphology of BA-MA were characterized, and a possible growth mechanism of the nanolayered structure was presented in detail. The resulting BA-MA with a thickness of about 60 nm can be uniformly dispersed in the PS resin, thus maintaining the mechanical properties of the material. Remarkably, under only 1 wt% loading of BA-MA, the flammability of PS can be largely reduced with a 68% reduction in the peak heat release rate. Additionally, the smoke release was also significantly inhibited. The research on flame retardant mechanisms shows that BA-MA mainly produces incombustible gas to dilute the concentration of combustibles and promote the formation of aromatic carbon layers to isolate oxygen transmission and heat transfer.

## 1. Introduction

Polymers are widely used in every part of our lives and play an irreplaceable role [1,2,3,4]. However, the vast majority of polymers are faced with high flammability, which limits their applications [5,6,7,8]. To meet flame safety requirements, many flame retardants have been employed to improve the flame retardancy of polymers, including halogen-free flame retardants and halogen-containing flame retardants that have been prohibited gradually [9,10,11,12].

Among halogen-free flame retardants [13,14,15], intumescent flame retardants (IFRs) have attracted much attention for their environmental friendliness and high flame retardant efficiency [16,17,18,19,20]. Generally, an intumescent flame retardant should contain three components: acid source, carbon source, and gas source [21,22,23,24]. A typical IFR system is ammonium polyphosphate (APP) and pentaerythritol (PER), which have been widely used to endow polymers with flame retardancy [25]. For instance, Xia et al. used ammonium polyphosphate (APP), pentaerythritol (PER), and melamine (MA) as an IFR system for the flame retardant treatment of polystyrene. When the flame retardant is 30 wt%, the limiting oxygen index of PS can reach 26.3%, and it can pass the UL-94 vertical burning test V-1 level [26]. Based on the IFR system, Yan et al. used a polymerized cyanuric chloride carbon-forming agent (CA) as a carbon source for flame retardant modification. When APP: CA = 3:1, the total amount of flame retardant was 30 wt%, the limiting oxygen index of the modified PS reached 32.5%, and the UL-94 vertical burning test passed the V-0 level [27]. “Three sources in one” is an important development direction of the intumescent flame retardant system. The acid source, carbon source, and gas source are concentrated on one substance to synthesize a single-added phosphorus-nitrogen synergistic flame retardant, which helps to improve the flame retardant efficiency. Tai et al. synthesized a P-N synergistic flame retardant PDMPD, and the charring performance of a PS material modified with PDMPD was improved [28,29].

However, APP has poor water resistance and tends to hydrolyze when it is exposed to a moist environment, which greatly decreases flame retardancy. In addition, the poor compatibility between the IFR and polymer matrix is inevitable, which would seriously deteriorate the mechanical properties of polymers. To solve these problems, a good solution is to produce polymeric IFRs [30]. It is a better way to introduce organic groups into polymeric IFR molecules, which will improve the compatibility of the IFR in the polymer matrix. On the other hand, the efficiency of flame retardants is affected by the aggregation structure and size of their particles [31,32], and good dispersion of flame retardants with nanostructures imparts the materials with excellent properties, including improved mechanical properties [33,34,35]. Biphenyl-4,4′-diphosphonic acid (BA) is composed of an aromatic group and two phosphonic acid groups and can properly act as an acid source and carbon source. BA is an organic acid, which would be beneficial to the dispersion property of the BA derivatives in the polymer matrix, and its chemical structure is convenient for assembling the nanolayered structure. Melamine (MA) is widely used as a fire retardant, especially in intumescent systems [36,37]. It is known that melamine can produce many nonflammable nitrogen-containing compounds, such as NH_3_, N_2,_ and NO_2_, during combustion, which can dilute combustible volatiles [38]. Thus, integrating the advantages of BA and MA into one molecule may be a promising candidate for intumescent flame retardants [39,40,41].

Based on the inspiration of the above-mentioned discussion, a unified organic nanosheet (BA-MA) is first designed and prepared. Due to the interactions between BA and MA, including electrostatic attraction, hydrogen bond, and π-π stacking, BA and MA are spontaneous to form an ordered aggregation, which has a hyperbranched nanolayered structure and possesses an acid source, a carbon source, and a gas source simultaneously. The chemical structure and morphology of BA-MA were characterized by FTIR, XPS, EDX mapping, SEM, and TEM, and a possible growth mechanism of the nanolayered structure is presented. The thermal stability, fire retardancy, and flame retardant mechanism of BA-MA in polystyrene (PS) resin were comprehensively investigated.

## 2. Experimental Section

### 2.1. Materials

Biphenyl-4,4′-diphosphonic acid (97%) was obtained from Hwrkchemical Company Limited (Beijing, China). Melamine was purchased from Kelong Chemical Reagent Factory (Chengdu, China) and used without further purification. Polystyrene resin (GP5250, industrial grade) was provided by Formosa Chemicals & Fibre Corporation (Taipei, China).

### 2.2. Preparation of Nano-Sheeted BA-MA

As shown in Figure 1, nano-sheeted BA-MA was synthesized employing the hydrothermal method by controlling the molar ratio, reaction temperature, and time duration. Briefly, 1.89 g (0.006 mol) of biphenyl-4,4′-diphosphonic acid and 0.51 g (0.004 mol) of melamine were added to a beaker, and 50 mL of deionized water was added. Then, a homogeneous mixture was obtained by using the ultrasonic dispersion method for half an hour. Subsequently, the mixture was totally transferred into a 100 mL Teflon-lined autoclave with the help of another 10 mL of deionized water. The autoclave was sealed and heated to 80 °C for 48 h. After cooling, the final product was filtered by a sand core funnel and dried in a vacuum oven at 60 °C for 12 h to a constant weight. The yield of the product was 91%, and the elemental analysis was carried out in Table 1.

### 2.3. Preparation of In Situ Dispersed PS/BA-MA

Suspension polymerization was employed to prepare the in situ dispersed PS/BA-MA. First, 30 mL of styrene monomer was washed with an equal volume of 10% NaOH solution to remove the polymerization inhibitor in the monomer and then washed with deionized water three times and transferred to a conical flask, and 0.30 g of freshly prepared BA-MA nanosheets was added. Then, 100 mL of deionized water was added into a three-necked round-bottomed flask with a reflux device with 0.30 g of PVA-1799 as a suspending agent. After heating to 90 °C, the St dispersion was added. A total of 0.10 g of the initiator was incorporated when a stable suspension system formed. The reaction was carried out for 8 h. The product was filtered, washed with water, and dried to obtain PS beads with in situ dispersed BA-MA nanosheets.

### 2.4. PS Blend Preparation

A series of PS blends with different loadings of BA-MA were prepared. In processing, PS resin and BA-MA with different mass fractions were fed into the internal torque rheometer (Kechuang XSS300, Shanghai, China) at 160 °C with a rotation speed of 100 rpm [14] for 7–10 min until the torque became stable.

### 2.5. Material Characterization

Fourier transform infrared spectra (FT-IR) were obtained on a Thermo Nicolet 6700 (Waltham, MA, USA) with a scanning range from 4000 to 400 cm^−1^. X-ray diffraction (XRD) was performed on a LabX XRD-6100 (Shimadzu, Kyoto, Japan) using Cu-Ka as the irradiation source (λ = 0.154178 nm, 5°/min). X-ray photoelectron spectroscopy (XPS) was performed on an XSAM 800 spectrometer (Kratos Co., Manchester, UK) using Al Kα rays (1486.6 eV) as the excitation source, and the working voltage and current were 12 kV and 15 mA, respectively. Scanning electron microscopy (SEM, Nova 600i, FEI, Hillsboro, OR, USA) and energy-dispersive X-ray spectroscopy (EDX) were used to observe the surface morphology and elemental analysis of the samples. A transmission electron microscope (TEM, Tecnai G2 F20 S-TWIN, 200 kV, FEI, Hillsboro, OR, USA) was used to characterize the morphology and microstructures of BA-MA and its dispersion in PS blends. Thermogravimetric analysis was performed on a Netzsch thermal analyzer (TGA, 209 F1, Serb, Germany) in a nitrogen atmosphere at a heating rate of 10 °C/min. The decomposition gases were identified by a thermogravimetry-infrared spectrometer (TG-IR), which was constructed by connecting the TA TGA5500 (Waters, Milford, MA, USA) thermogravimetric analyzer to the Nicolet iS50 FT-IR spectrophotometer (Thermo, Waltham, MA, USA). Samples (10 ± 0.5 mg) were measured from 40 °C to 700 °C at a 20 °C/min heating rate in a nitrogen atmosphere (flow rate: 25 mL/min). The flammability of the sample was investigated by using a cone calorimeter (Fire Testing Technology, West Sussex, UK) according to ISO 5660. The samples with 100 × 100 × 3 mm^3^ were irradiated at a heat flux of 35 kW/m^2^. The measurements were performed in triplicate, and the average data were reported.

## 3. Results and Discussion

### 3.1. Fabrication and Morphology

First, a unified organic flame retardant (BA-MA) with a hyperbranched nanolayered structure was prepared from biphenyl-4,4′-diphosphonic acid and melamine. Then, it was blended with the PS substrate to obtain a flame retardant PS material. It is worth noting that BA contains two aromatic fragments and two phosphonic acid groups, which can play a good role as acid and carbon sources. Melamine, as a gas source, has successfully realized the “Trinity” intumescent flame retardant. After exploring the preliminary experiment, we chose the PS blends containing 1 wt% and 2 wt% of BA-MA for in-depth performance research and discussion. The obtained flame retardant sample avoids the problem of easy hydrolysis of the traditional intumescent flame retardant APP. In addition, the flame retardant is nanolayered, which has better compatibility with the polymer matrix.

To investigate the chemical structure of BA-MA, the FTIR spectra of BA, MA, and BA-MA were examined, as shown in Figure 1. For MA, the multiple absorption peaks of 3100–3500 cm^−1^ belong to -NH_2_, and the number of peaks is reduced to two in the FTIR spectrum of BA-MA. The two relatively broad peaks at 3383 and 3120 cm^−1^ are related to the stretching vibration absorption of -NH_3_^+^ and unreacted -NH_2_. The stretching vibration absorption of -NH_2_ is 3133 cm^−1^ in MA. That is the result of hydrogen bonds between –NH_2_ (from MA) and P=O (from BA). The bending vibration of NH_2_ in MA appears at 1648 cm^−1^_,_ while in BA-MA, it shifts to 1668 cm^−1^ due to the intermolecular interactions through the -NH_2_ groups. In comparison, a new absorption band, appearing at 1510 cm^−1^ for BA-MA, can be ascribed to -NH_3_^+^. For BA, there is a broad and strong peak that can be classified as a stretching vibration of the -PO_3_H_2_ associating system, which disappeared in BA-MA for the destroyed system. These characteristic absorption peaks reveal that a neutralization reaction does occur between BA and MA.

To further confirm the chemical structure of BA-MA, XPS tests were carried out, and the corresponding spectra are shown in Figure 2. For BA-MA, the peaks located at 132.1, 190.1, 399.0, and 531.1 eV are assigned to P_2P_, P_2S_, N_1s,_ and O_1S_, respectively. In contrast to BA and MA, the binding energy of P_2P_ and N_1s_ changed. The fitted spectra of P_2P_ and N_1s_ are shown in Figure 2b–d. In the N_1s_ spectra of MA (Figure 2b), the peak at 399.6 eV corresponds to -NH_2_, and the peak at 398.7 eV is due to -C=N-. In the N_1s_ spectra of BA-MA (Figure 2c), the peaks of -NH_2_ and -C=N- are similar to those in MA with a slight shift of binding energy, except for a new peak at 400.3 eV, which should be assigned to -NH_3_^+^. Regarding the P_2p_ spectra of BA-MA shown in Figure 2d, the peak at 134.1 eV and the peak at 133.3 eV can be assigned to H_2_PO_4_^−^ and HPO_4_^2−^, respectively [42]. All the results demonstrate that the hyperbranched BA-MA molecules have been synthesized successfully.

As we know, changes in chemical structure lead to changes in the crystal structure. Therefore, the crystal structure of BA-MA was investigated by X-ray diffraction. Figure 3 shows the XRD patterns of BA, MA, and BA-MA, respectively. Compared with the MA pattern, the characteristic peaks disappear in the BA-MA pattern, which can be attributed to the high degree of hybridization between BA and MA, leading to the destruction of the inherent crystal structure of MA and forming a new state of aggregation. On the other hand, the characteristic peaks of the BA pattern changed after reacting with MA due to the π-π stacking interactions between aromatic rings in BA-MA, which led to a change in interlayer spacing. Relying on the Bragg equation (2dsinθ = nλ), the interlayer spacing of BA (2θ = 19.94°) is 0.445 nm, that of MA (2θ = 26.12°) is 0.341 nm, and that of BA-MA (2θ = 19.16°) is 0.464 nm. After the reaction between BA and MA, the interactions between phosphonic acid sites were destroyed.

As shown in Figure 4a–c, the morphology of the BA-MA sample was characterized by SEM. It can be seen that BA-MA exhibits a nanosheet structure with a thickness of approximately 60 nm. In addition, a large number of nanosheets were aggregated into microspheres with a size of 50 μm. In the hydrothermal reaction of BA and MA, the two first combine to form BA-MA nanosheets, and then the nanosheets aggregate into larger microspheres due to the equilibrium between electrostatic attraction and interlayer repulsion. On the one hand, this nanosheet structure is conducive to its dispersion in PS; on the other hand, it may play a nano-enhancing role and reduce the decrease in PS mechanical properties caused by its addition. Furthermore, combined with the TEM image of the PS bead profile of the in situ dispersed BA-MA nanosheets, it can be seen that the linear gaps on the PS bead profile are uniformly distributed. Additionally, their size is consistent with the thickness of the BA-MA nanosheets, indicating that the in situ dispersion is achieved during the preparation. BA-MA nanosheets are uniformly dispersed in PS beads, and no agglomeration occurs.

### 3.2. Thermal Stability

The TG and DTG curves of BA-MA nanosheets, pure PS, and PS/BA-MA blends are shown in Figure 5a,b. The initial thermal decomposition temperature (*T*_5%_) of BA-MA nanosheets is 307 °C, which is lower than the 366 °C of the PS pure sample. The maximum thermal decomposition temperature *T_max_* of BA-MA is 423 °C, which is higher than 415 °C of pure PS. The maximum thermal degradation rate of PS blended with 2% BA-MA nanosheets is −22.53%/min, significantly lower than that of the pure sample −32.60%/min, indicating that adding BA-MA can effectively slow down the thermal decomposition rate of PS. Thus, the production rate of small molecules can be reduced at high temperatures, which has a positive effect on flame retardancy. The char residue of BA-MA is 48% at 600 °C, which can isolate flame, heat transfer, and oxygen, inhibit the volatilization of combustible gas, so as to play a good role in flame retardancy.

Further, to study the effect of adding BA-MA on the thermal decomposition behavior of PS, by assuming that PS and BA-MA do not have any interaction during the decomposition process, we obtained the fitting curves according to the content of BA-MA in the blend material [27]. The theoretical TG and DTG curves of the material are shown in Figure 6a,b. The above-detailed data are shown in Table 2. The fitting curve and the actual test curve do not overlap, and the maximum thermal decomposition temperature in the fitting curve is 416 °C lower than the actual curve value, which proves that BA-MA can inhibit the decomposition of PS under actual conditions. The maximum thermal degradation rate of the curve is −31.99%/min, much higher than the −22.53%/min of the actual curve. It is suggested that there is an interaction between BA-MA and PS in the blended sample during the decomposition process, which inhibits the degradation of PS at high temperatures.

### 3.3. Flammability

To further characterize the flame retardant properties of BA-MA and its effect on the combustion behavior of PS, PS samples with BA-MA additions of 1 wt% and 2 wt% were tested by cone calorimetry and compared with pure PS [43,44,45,46]. Relevant data are shown in Figure 7 and Figure 8, and Table 3. The ignition time of unmodified PS was 55 s, while the ignition time of PS samples modified with 1 wt% and 2 wt% BA-MA was reduced to 51 s and 45 s, respectively. Figure 7 shows the heat release rate curves and total heat release curves of the samples. The peak heat release rate of unmodified PS was 702.38 kW/m^2^, and the total heat release was 113.06 MJ/m^2^. After adding only 1 wt% or 2 wt% BA-MA, the peak heat release rate of the samples decreased to 593.97 and 563.45 kW/m^2^, respectively. The fire growth rate decreased from 3.5 kW/m^2^/s to 2.9 and 2.5 kW/m^2^/s, respectively. This shows that the aromatized char layer generated by the decomposition of BA-MA at the early stage of combustion inhibits the intensity of material combustion and delays the combustion of PS. The total heat release was also slightly inhibited.

Figure 8 shows the smoke production rate curves and the total smoke production curves. The peak smoke production rate of the unmodified PS sample was 0.22 m^2^/s, and the total smoke production was 33.96 m^2^/kg. With the addition of BA-MA-modified PS samples, the peak smoke production rate decreased, but the total smoke production was almost unchanged. The results of total smoke production indicated that the addition of BA-MA could not inhibit the total amount of smoke produced by PS during combustion. The reduction in the peak smoke production rate should be attributed to the aromatized char layer generated by the decomposition of BA-MA at the early stage of combustion, which blocked the fragments generated during decomposition. However, in the further burning process of the material, the aromatized char layer of BA-MA is destroyed, and the previously blocked molecular fragments participate in the combustion again. Therefore, the heat release rate curves and the smoke production rate curves showed a sustained release platform.

In the cone calorimetry test, the destruction of the char layer of BA-MA during combustion would lead to a decline in the flame retardant properties. Therefore, a microcalorimeter (MCC) was used to study the performance of the samples in small-scale flammability tests. The results showed that with the addition of 1 wt% BA-MA, the peak heat release rate of the sample was reduced from 1286 W/g to 406 W/g, with a decrease of 68%, and the heat capacity and total heat release of the material are also dropped. That is, when the char layer generated by BA-MA during combustion is strong enough without being destroyed, it can exhibit excellent flame retardant properties. Detailed data are shown in Figure 9 and Table 4.

### 3.4. Flame Retardant and Smoke Suppression Mechanism

TG-FTIR analysis can be used to conduct real-time infrared characterization of the gaseous products generated by the degradation of the material during the thermal degradation process to determine the components and clarify the thermal decomposition behavior of the material. Thus, TG-FTIR tests were performed on BA-MA, PS, and PS/BA-MA02. The FTIR 3D map of the evolved products at various times and typical FTIR spectra at different temperatures during the thermal degradation of BA-MA, PS, and PS/BA-MA02 are shown in Figure 10, Figure 11, and Figure 12, respectively. Due to the high carbonization rate of BA-MA at high temperatures, its peak intensity is low, mainly including the NH stretching vibration peak at 3300 cm^−1^, the PH stretching vibration peak at 2320 cm^−1^, and the CN stretching vibration peak at 1436 cm^−1^.

For pure PS, the peaks mainly include the stretching vibration peak of =CH at 3050 cm^−1^, the stretching vibration peak of C=C at 1640 cm^−1^, the skeleton vibration peak of the benzene ring in the range of 1650–1450 cm^−1^, and the skeleton vibration peak of 750 cm^−1^. Its peak position is the same as that of the styrene monomer, indicating that PS mainly decomposes to the styrene monomer and its oligomers in the process of thermal degradation.

In contrast, the peak positions of PS/BA-MA02 mainly include 3070 cm^−1^ olefinic bond and the stretching vibration peak of CH on the benzene ring, the C=C stretching vibration peak at 1640 cm^−1^, and the out-of-plane bending vibration peak of benzene ring CH at 750 cm^−1^. The relative absorption peaks of BA-MA degradation products were not obvious. This shows that, in the process of thermal degradation of PS/BA-MA02, the typical thermal degradation behavior of PS is dominant. Compared with the TG-IR spectrum of pure PS, the relative intensities of each absorption peak of PS/BA-MA02 are reduced, indicating that the introduction of BA-MA can effectively inhibit the thermal degradation of PS.

To better study the degradation behavior of BA-MA, the char residue generated in a tube furnace under different temperatures was characterized by FT-IR. The results showed that after decomposition in the tube furnace, the BA-MA nanosheets mainly remained in the composite char layer mainly composed of triazine ring and benzene ring structures. As shown in Figure 13, combined with the TG-IR results, it can be seen that BA-MA nanosheets are mainly decomposed to generate PH_3_ and NH_3_ during the thermal degradation process, and the residual triazine and benzene ring form an expanded char layer for the production of gases [47]. The flame retardant mechanism of BA-MA is summarized. For the PS/BA-MA system, when the material burns in a fire, BA-MA decomposes to generate PH_3_ and NH_3_ and form a char layer dominated by triazine and benzene rings. The char layer expands to block oxygen and heat transfer, preventing the material from further burning [48].

### 3.5. Mechanical Property

The tensile and flexural strengths of the flame retardant-modified PS samples were tested and compared with those of the pure PS samples. The detailed mechanical property test data are shown in Table 5. After adding BA-MA, the tensile strength and flexural strength of the PS samples did not change significantly. When the sheet was introduced into the PS matrix by blending, the compatibility with PS was good enough, which was the main reason that the mechanical strength of the composite can remain. In addition, the π-π stacking structure of BA-MA nanosheets can also reduce the possibility of brittle fracture of the blended material when subjected to external force.

## 4. Conclusions

In summary, we developed a unique organic nanolayered intumescent flame retardant BA-MA. Due to the good dispersion between BA-MA and PS, the mechanical properties of the flame-retardant PS composites can be maintained. PS-02 also effectively inhibited the high-temperature thermal degradation of PS, i.e., its maximum weight loss rate of the material decreased by 31%. Further, in the cone calorimetry test, the PHRR of the material decreased by 20% after adding 2 wt% BA-MA. Similarly, in the MCC test, the PHRR of PS with 1 wt% BA-MA decreased by 68%. Therefore, the synthesized flame retardant with nanolayered structures shows excellent heat release inhibition performance at low addition. The study of the flame-retardant mechanism showed that BA-MA mainly produced flame-retardant gas to dilute the concentration of combustibles, and further promoted the formation of aromatic carbon layers, which acted as oxygen and heat insulation. This study provides a new idea for the synthesis of flame retardants with nanolayered structures compatible with the corresponding polymer matrix.

## Data Availability

The data presented in this study are openly available with suitable citations.

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
