# Peer review of "A Phosphorus-Nitrogen-Carbon Synergistic Nanolayered Flame Retardant for Polystyrene"

_polymers, 2022, doi:10.3390/polym14102055_

Round 1

Reviewer 1 Report

In this work  an assembly of biphenyl-4,4'-diphosphonic acid (BA) and melamine (MA) was used as an additive flame retardant for polystyrene. It has been found that at 1wt% loading of BA-MA,  the flammability of PS was reduced by  68%, and the smoke release was also significantly suppressed.  BA-MA mode of action in gas phase leads to dilution of the concentration of combustible gases, as well as promotes the formation of aromatic carbon barier layers. The concept presented is interesting, however, there are several points that need to be addressed in more depth:

- The chemical reaction between melamine and phenyl-4,4'-diphosphonic acid is unlikely to happen at these reaction conditions. And Scheme 1 suggests such a reaction that has not been evidenced by any spectroscopic method. 

-  what you mean by „the assembly of combining bi-phenyl-4,4'-diphosphonic acid with melamine” (Lines 71-72)?  What is the structure of this assembly? How it was identified?

- In phenyl-4,4'-diphosphonic acid, the periodic organization of the molecules is controlled by two strong O-H...O interactions between the phosphonic acid sites [e.g. Acta Crystallogr C. 2007 Jul; 63:434-6].  Please check these interactions by IR and WAXD methods.

-  What about hydrogen bonds between –NH2 (from MA) and P=O (from BA)?

- How the processing parameters (Sub-section 2.4) were selected?

- Fig. 5 – please discuss the (large) char residue effect of BA-MA vs flame retardancy.

-  Conclusions need to be re-written – please avoid abstract-like style.

English needs to be checked, e.g. in the title „A P-N-C unified organic…”.     

Reviewer 2 Report

The manuscript 'A P-N-C unified organic nanolayered flame retardant for polystyrene' is devoted to the development of efficient flame retardants compatible with PS.  The manuscript  clearly creates a good impression, meets high level of the Polymers journal by the criteria of scientific relevance and quality of presentation, and therefore can be published after minor revision.

Comments and recommendations

  1. Section 2.1 (and below) – please specify city, (state) and country for the manufacturers of chemicals and equipment. Please use hyphens, dashes and minus signs correctly. Please use italic font for variables. Don't use bold font to highlight figures and schemes in the text.
  2. Section 2.2 – the yield of the product (g, %) and elemental analysis data are needed. The Scheme 1 should be redrawn using common template (for example, ACS)
  3. Section 3.1

- line 144 – strictly speaking, BA contains two aromatic fragments

- Fig. 2 – the quality of this Figure should be improved

  1. Section 3.4, Figs. 10–12 – please enlarge

References section should be formatted according to Polymers' template

Round 2

Reviewer 1 Report

Authors have revised their work, and the revised manuscript can be published.